# Cuproptosis-Related lncRNA Gene Signature Establishes a Prognostic Model of Gastric Adenocarcinoma and Evaluate the Effect of Antineoplastic Drugs

**DOI:** 10.3390/genes13122214

**Published:** 2022-11-25

**Authors:** Hengjia Tu, Qingling Zhang, Lingna Xue, Junrong Bao

**Affiliations:** 1Nanshan School, Guangzhou Medical University, Guangzhou 511436, China; 2The First Affiliated Hospital of Guangzhou Medical University, Guangzhou 510120, China; 3Faculty of Big Data and Computing, Guangdong Baiyun University, No.1 Xueyuan Road, Guangzhou 510450, China

**Keywords:** Cuproptosis, Gastric Adenocarcinoma, STAD, lncRNA, prognostic, antineoplastic drugs

## Abstract

Background: One of the most frequent malignancies of the digestive system is stomach adenocarcinoma (STAD). Recent research has demonstrated how cuproptosis (copper-dependent cell death) differs from other cell death mechanisms that were previously understood. Cuproptosis regulation in tumor cells could be a brand-new treatment strategy. Our goal was to create a cuproptosis-related lncRNA signature. Additionally, in order to evaluate the possible immunotherapeutic advantages and drug sensitivity, we attempted to study the association between these lncRNAs and the tumor immune microenvironment of STAD tumors. Methods: The TCGA database was accessed to download the RNA sequencing data, genetic mutations, and clinical profiles for TCGA STAD. To locate lncRNAs related to cuproptosis and build risk-prognosis models, three techniques were used: co-expression network analysis, Cox-regression techniques, and LASSO techniques. Additionally, an integrated methodology was used to validate the models’ predictive capabilities. Then, using GO and KEGG analysis, we discovered the variations in biological functions between each group. The link between the risk score and various medications for STAD treatment was estimated using the tumor mutational load (TMB) and tumor immune dysfunction and rejection (TIDE) scores. Result: We gathered 22 genes linked to cuproptosis based on the prior literature. Six lncRNAs related to cuproptosis were used to create a prognostic marker (AC016394.2, AC023511.1, AC147067.2, AL590705.3, HAGLR, and LINC01094). After that, the patients were split into high-risk and low-risk groups. A statistically significant difference in overall survival between the two groups was visible in the survival curves. The risk score was demonstrated to be an independent factor affecting the prognosis by both univariate and multivariate Cox regression analysis. Different risk scores were substantially related to the various immunological states of STAD patients, as further evidenced by immune cell infiltration and ssGSEA analysis. The two groups had differing burdens of tumor mutations. In addition, immunotherapy was more effective for STAD patients in the high-risk group than in the low-risk group, and risk scores for STAD were substantially connected with medication sensitivity. Conclusions: We discovered a marker for six cuproptosis-associated lncRNAs linked to STAD as prognostic predictors, which may be useful biomarkers for risk stratification, evaluation of possible immunotherapy, and assessment of treatment sensitivity for STAD.

## 1. Introduction

One of the biggest causes of cancer-related fatalities globally is gastric cancer (GC) [1]. The most prevalent subtype of GC is understood to be stomach adenocarcinoma (STAD). At the time of initial diagnosis, most STAD patients had distant metastases. Significant advancements in the treatment of advanced STAD, including targeted treatments and immunotherapy as well as conventional chemotherapy, have improved prognosis in recent years [2]. Although STAD is a highly diverse tumor, many patients in clinical practice do not benefit from immunotherapy or targeted therapy [3]. Therefore, it is essential to choose those patients who might benefit from these treatments.

Long non-coding RNAs (lncRNAs) are non-coding functional RNAs. Numerous lncRNAs are related to patient prognosis in STAD [4,5,6], even though several lncRNAs have been linked to the prognosis of gastric cancer and may be exploited as prognostic markers in future prognostic studies. However, because lncRNAs and genes may be associated, unilateral examination of lncRNA is frequently unstable. Therefore, a model incorporating various lncRNAs with predictive impact must be created. A nutrient called copper plays a role in the pathways that control cell growth and degeneration. Peter Tsvetkov et al. [7] recently described a novel type of cell death called cuproptosis, which is distinct from the known processes of death such apoptosis, pyroptosis, necrosis, and ferroptosis. Cuproptosis has been shown to be associated with the occurrence and development of a variety of cancers, including liver cancer [8], colorectal cancer [9], esophageal cancer [10], and some hematologic malignancies, including the acute myeloid leukemia [11]. These studies suggest that cuproptosis may be associated with a wider range of malignancies. However, the relationship between cuproptosis and STAD remains unclear. The lncRNAs related to cuproptosis may serve as a potential diagnostic, typing and therapeutic targets for STAD in the future.

To conduct a comprehensive analysis of cuproptosis-related gene (CRG) expression, mutational status, and copy number alterations, we gathered STAD samples from The Cancer Genome Atlas (TCGA) database. To fully assess the relationship between various risk strata and the tumor immune microenvironment (TIME), predictive models were created. Additionally, we looked into how sensitive patients were to STAD immunotherapeutic medicines based on their risk levels. Our research offers fresh perspectives on STAD categorization and effective therapy.

## 2. Method

### 2.1. Data Collection

The TCGA database (https://cancergenome.nih.gov/ (accessed on 11 September 2022) provided the RNA sequencing data as well as pertinent clinical and follow-up data. Stomach adenocarcinoma (STAD) FPKM values were converted to transcripts per kilobase million and then normalized (TPM). We combined the two cohorts and eliminated patients whose survival status was uncertain or for whom there was insufficient follow-up data. In this study, 371 STAD patients were enrolled for analysis. Clinical information was gathered, including TNM stage, pathological grade, age, gender, follow-up period, and survival status. In this study, 22 genes associated with cuproptosis were chosen for examination based on previously published studies [7,12,13,14].

### 2.2. Construction of Cuproptosis-Related Prognostic Signature for STAD

Cuproptosis-related lncRNAs that were strongly correlated with survival were identified using LASSO regression and Univariate Cox regression analysis. A 6-lncRNA signature was created based on the regression coefficient and prognostic potential.

### 2.3. Establishment and Evaluation of the 6-lncRNA Signature

All of the patients were separated into two groups at random, one of which served as the test set and the other as the train set. The median risk score is utilized in the train set to categorize patients into two categories (high and low risk, respectively). The R software packages “survival” and “survminer” were used to perform the Kaplan–Meier survival analysis. Additionally, the accuracy and diagnostic value of the 6-lncRNA signature were assessed using the receiver operating characteristic (ROC) curve and the area under the curve (AUC). The consistency of these results in the test set was subsequently confirmed. In order to validate risk models, the principal component analysis (PCA) was also carried out [15]. Scatterplot3D programs in R software were used to display the results. The R “survival” and “survminer” packages were used to calculate the progression-free survival (PFS). We utilized the R packages “rms”, “dplyr”, “survival”, and “pec” to predict the accuracy of risk models using the C-index.

### 2.4. Exploration of the Relationship between the Prognostic Risk Score and Clinical Stage

We investigated the correlations between risk score and clinical stage using univariate and multivariate Cox regression analyses to determine whether the model is appropriate for patients with various clinical stages.

### 2.5. Construction of Nomogram

We developed nomograms for predicting survival in STAD patients using the clinical parameters, such as TNM stage, age, and gender. Calibration curves were also created in order to evaluate the consistency between anticipated and actual survival.

### 2.6. Enrichment Functional Analysis

We started by comparing the expression of a group of DEGs, or differentially expressed genes, in the two groups. We used the following rules by referring to previous studies [8,16]: if the LogFC (fold change) is greater than 1 or less than −1 (|logFC| > 1), we think that the gene is expressed significantly differently in the two risk groups. Furthermore, the FDR (false discovery rate) should less than 0.05. Following that, KEGG pathway analysis and GO function enrichment analysis were carried out for selected DEGs.

### 2.7. Estimation of Intratumoral Immune Cell Infiltration

We measured the number of immune cells in the two risk groups using the ssGSEA algorithm. In addition, according to previous research, the clinical effectiveness of immune checkpoint inhibitor blocking therapy may be connected with the level of gene expression in immune checkpoint-related tissues. Thus, it was also investigated whether there was a relationship between risk scores and immunological checkpoints.

### 2.8. Estimation of Tumor Mutation Burden

The quantity of tumor mutations is reflected by the tumor mutational burden (TMB). Using the R package “maftools” the mutation data of HNSC samples acquired from TCGA were examined. The waterfall diagram demonstrated the connection between TMB and risk ratings in HNSC patients. The immunological reaction was forecasted using the tumor immune dysfunction and exclusion (TIDE) method.

### 2.9. Evaluation of Drug Sensitivity

The measured antagonist’s measured IC50 concentration was its semi-inhibitory level. We used the “pRRophetic” R package and its dependencies “car, ridge preprocess Core, genefilter, and sva” to determine the IC50 of the chemotherapeutic medicines in order to evaluate CRLPM in the clinic for the treatment of HNSC. There were 19 different medications total. The IC50 discrepancies between popular antineoplastic drugs in the high- and low-risk categories were compared using the Wilcoxon sign rank test. The “ggplot2” R package was used to present the boxplot.

### 2.10. Statistical Analysis

R software (version 4.0.3; available at https://www.r-project.org/ (accessed on 11 September 2022) and Perl (version 5.32.1.1; available at https://strawberryperl.com/ (accessed on 11 September 2022) were used to perform Pearson’s correlation, Cox regression, and Kaplan–Meier curve analyses. Differences with *p*-values of 0.05 or above are statistically significant.

## 3. Results

### 3.1. Data Processing

A total of 371 STAD samples were randomly assigned to the test set (*n* = 185) or the training set (*n* = 186). The clinical characteristics of the patients in the two sets are shown in Table 1. The differences among all the baseline variables were not statistically significant (all *p*-values > 0.05).

We identified 17648 lncRNAs in the TCGA_STAD dataset. In total, we collected 22 cuproptosis-related genes. The Sankey plot showed the association between cuproptosis-related genes and cuproptosis-related lncRNAs (Figure 1).

### 3.2. A Cuproptosis-Related Long Noncoding RNAs Prognostic Marker for STAD: Development and Validation

The “glmnet” package of R software was used to find the cuproptosis-related lncRNAs with the highest prognostic values. We identified sixteen lncRNAs (Figure 2A). Among them, 14 lncRNAs with a hazard ratio (HR) > 1, LINC01094, AC147067.2, AC037198.1, MSC−AS1, AC110611.2, AL359704.2, HAGLR, AC006033.2, AL590705.3, AC133106.1, AL139147.1, AL670729.3, VCAN−AS1, and AC023511.1 were found to be poor prognostic predictors, while the other two, PINK1−AS and AC016394.2, may be protective indicators. We then evaluated six lncRNAs using LASSO (least absolute shrinkage and selection operator) regression (Figure 2B,C). The correlation heatmap demonstrated the relationship between six tested lncRNAs and genes involved to cuproptosis (Figure 2D). Risk score = (−0.6558*AC016394.2 expression) + (0.4407*AC023511.1 expression) + (0.3219*AC147067.2 expression) + (0.7508*AL590705.3 expression) + (0.2162*HAGLR expression) + (0.8137*LINC01094 expression) was the formula used to create the risk score model.

The patients in the test group and training group were then split into high- and low-risk groups for survival analysis based on the median risk score. The OS of patients in the two groups was analyzed using the KM technique (Figure 3A–C). The overall survival rate (OS) in both groups was substantially greater in the high-risk group than in the low-risk group (*p* = 0.05) The cuproptosis-related lncRNAs’ diagnostic usefulness for the OS of STAD patients was demonstrated by the ROC curves (Figure 3D). For the 1-, 3-, and 5-year ROCs, the area under the curve (AUC) was 0.703, 0.722, and 0.680, respectively. Additionally, the risk score (AUC = 0.703) has a stronger diagnostic value when compared to other variables (Figure 3E), such as age (AUC = 0.594), gender (AUC = 0.522), tumor grade (AUC = 0.561), and tumor stage (AUC = 0.602). The C-Index curve (Figure 3F) showed that the risk model had higher consistency than other variables. Figure 4A–F displays the distribution of risk scores and the patients’ survival status. The survival time was decreased and the death rate rose in all three groups as the risk score rose. A heatmap depicts the expression levels of six lncRNAs related to cuproptosis that were active in three groups (Figure 4G–I).

### 3.3. Independent Prognostic Marker for Cuproptosis-Related lncRNAs in Predicting Overall Survival

To examine the predictive potential of the prognostic signature employing cuproptosis-related lncRNAs in STAD, univariate and multivariate Cox regression analyses were undertaken. Age, stage, and risk score all showed statistically significant differences in the univariate Cox analysis (Figure 5A). They continued to be statistically significant in the multivariate Cox regression analysis (Figure 5B). They show that the risk score can independently predict the prognosis of STAD patients (HR = 1.134, 95% CI = (1.074–1.197), *p =* 0.001). Between the high- and low-risk groups, there is a statistically significant difference in PFS (progression-free survival) (*p* = 0.001, Figure 6A). After that, we looked into the reasons behind the discrepancies between the high- and low-risk groups using PCA. Figure 6C,D shows that there are no variations in the expression of all genes, all 22 cuproptosis-related genes, or all 16 cuproptosis-related lncRNAs (Figure 6E). However, the six cuproptosis-associated lncRNAs were the most effective at differentiating between patients at low and high risk (Figure 6B). Patients were further separated into subgroups to examine whether clinical stages had an impact on the prognostic signature’s capacity to predict outcomes. All high-risk patients were discovered to have a worse prognosis in both stage I–II (Figure 7A) and stage III–IV (Figure 7B) categories. For the purpose of analyzing clinical outcomes in STAD patients quantitatively, a predictive nomogram model comprising eight variables was developed (Figure 7C). The calibration curves demonstrate that the model’s predictions and the measured values correspond quite well (Figure 7D). In conclusion, cuproptosis-related lncRNAs have independent and reliable prognostic prediction abilities.

### 3.4. Functional Enrichment Analysis

To learn more about the lncRNAs and mRNAs that were differentially expressed in the two risk groups, we used GO (Figure 8A,B) and KEGG (Figure 8C,D) pathway analysis. Skeletal system development and ossification were highly enriched in the BP (biological processes) category. The DEGs were primarily concentrated in collagen-containing as well as six other pathways within the CC (cellular components) category. DEGs were primarily enriched in receptor–ligand activity and signaling receptor activator activity in the MF (molecular functions) category. Differentially expressed genes were mainly enriched in the PI3K-Akt signaling network, neuroactive receptor–ligand interaction, and another 22 pathways, according to KEGG pathway analysis.

### 3.5. Estimation of Intratumoral Immune Cell Infiltration and Immunotherapy

Figure 9A depicts the results of our investigation into the relationship between risk ratings and immune-related activities in STAD. The heatmap demonstrates the striking differences between the CCR, Type 1 IFN response, HLA, APC co-stimulation, parainflammation, Type II IFN response T-cell co-inhibition, Cheak point, and T-cell co-stimulation in various risk classes. It is interesting to note that in high-risk patients, almost all immune-related processes are more active.

### 3.6. Tumor Mutational Burden of the Cuproptosis-related lncRNAs Prognostic Marker in STAD Samples

TMB (tumor mutation burden) was significantly different between the two groups in general (Figure 9B). It is unclear why the high TMB group demonstrated a better prognosis than the low TMB group (Figure 9C). The prognosis of STAD patients was then further evaluated by combining risk score and TMB, and we discovered that, regardless of the TMB level, the high-risk group had a poorer chance of survival than the low-risk group (Figure 9D). From the TGCA database, we obtained the somatic mutation data, and we compared the mutation rates between the two groups (Figure 9E,F). The findings revealed that the high-risk group had a greater mutation frequency than the low-risk group (90.26% vs. 89.16%). The two genes with the highest mutation frequency were *TTN* and *TP53*.

### 3.7. Drug Sensitivity

Analyses were performed on the immunotherapy’s sensitivity variations between the two risk groups. The low-risk group had considerably lower TIDE scores than the high-risk group (Figure 10A), indicating that they were more likely to receive effective immunotherapy and less likely to experience immunological escape. The half maximum IC50 (half maximum inhibitory concentration) of several medications was determined in two groups to reveal the relationship between drug sensitivity and risk scores in order to determine whether our risk score model might be used in the tailored treatment of STAD (Figure 10B–K). The IC50 of Gefitinib increased as the risk score rose, but it reduced for Cytarabine, Dasatinib, Pazopanib, and Saracatinib. Cytarabine, Dasatinib, Pazopanib, and Saracatinib had a greater IC50 for the low-risk group, but Gefitinib had a higher IC50 for high-risk group. Appendix A display the outcomes of the other 14 medications. In conclusion, the findings imply that our risk model might guide therapeutic care for STAD patients.

## 4. Discussion

One of the tumors of the digestive system that pose significant health hazards is STAD, which has drawn attention from people all over the world [8]. Studying prospective biomarkers and potentially tumor-promoting or suppressor genes [9,17] in STAD is therefore worthwhile.

The primary mechanism of tumor treatment resistance is tolerance to apoptosis, which is mainly induced by conventional chemotherapeutic agents in tumor cells [18]. Comprehensive research on the apoptotic pathways in tumor cells in recent years has gradually revealed new types of programmed cell death, including pyroptosis, ferroptosis, and necroptosis. Most significantly, cuproptosis is the most recent type of programmed cell death to be described. Lipid peroxidation and significant iron accumulation are frequently present in addition to it [19,20]. A loss in cellular antioxidant capability, a buildup of lipid reactive oxygen species (ROS), and ultimately oxidative cell death can result from the direct or indirect effects of iron atrophy inducers on glutathione peroxidase [21]. Similarly, copper can cause cell death by increasing energy metabolism in the mitochondria and ultimately oxidative cell death.

Similar to the way it causes cuproptosis, copper can also cause cell death via increasing energy metabolism that is dependent on the mitochondria and cytotoxicity brought on by a buildup of ROS. According to research, copper is a dynamic signaling metal and an extraterrestrial bioregulator that controls and coordinates biological activity in response to environmental cues. When the copper transporter CTR1 or ULK1 is genetically lost or mutated, copper binding is disrupted, which lowers ULK1/2-dependent signaling, autophagosome complex formation, and lung cancer development and survival [22]. The RAS/RAF/MEK/ERK (MAPK) signaling cascade controls fundamental cellular processes such as cell proliferation, survival, and differentiation. It is also crucial for intracellular communication. Increased levels of activation upstream of receptor tyrosine kinase TRKB, EGFR, and MAPK signaling were seen in copper-treated cancer cells [23,24]. The E2-binding enzymes UBE2D1 and UBE2D4 are heterologously activated by copper metallization to accelerate protein breakdown. As a result, ubiquitin labels and destroys several proteins, most notably *p53*. Therefore, the overabundance of copper in malignant cells causes the attenuation of *p53*, which may contribute to the tumor cells’ resistance to programmed cell death [25].

In this study, to gain insight into cuproptosis-associated lncRNAs with STAD prognosis, we first downloaded the expression profiles of lncRNAs and genes in STAD patients provided by the TCGA database. By cox regression and LASSO analysis, we identified six prognosis-related lncRNAs: AC016394.2, AC023511.1, AC147067.2, AL590705.3, HAGLR, and LINC01094. we constructed a 6-lncRNA signature risk score model. We further divided the patients into high-risk score and low-risk score groups based on median risk scores. Patients with high-risk scores had significantly shorter survival. ROC analysis was used to evaluate the accuracy of this lncRNA risk model. the high AUC value at 5 years (AUC = 0.608) suggested that the risk model was reliable for the prognosis of STAD. The risk score (AUC = 0.703) had a better diagnostic value than other variables. The Cox regression analysis of the risk score model and clinical characteristics suggested that the genetic risk model could be used as an independent predictor for prognostic evaluation of STAD.

In addition, we evaluated the tumor immune microenvironment of STAD patients in different risk groups, and almost all immune-related activities were more active in high-risk patients. We evaluated the TMB in STAD patients, and we found that *TTN* and *TP53* were the two genes with the highest mutation frequencies, and the probability of 3-, 5-, and 10-year survival was lower in the high-risk group than in the low-risk group, regardless of TMB levels. Finally, we evaluated 19 STAD therapeutic agents and found different “risk score-sensitivity” relationships. The results suggest that our risk model can inform the clinical treatment of STAD patients.

However, our work has some limitations. Firstly, we did not demonstrate our results in in vitro tissues. Secondly, some novel lncRNAs with clinical importance in STAD need to be further explored to determine their potential molecular mechanisms. Thirdly, the training and test sets were equally separated in this article. Although the statistical consistency of clinical indicators can be achieved by reducing the sample difference between two groups, the performance of the parameter estimation through the training set may degrade.

In conclusion, we identified a practical prognostic model based on cuproptosis-associated lncRNAs in STAD and analyzed the relationship between cuproptosis and tumor immune correlates. Our findings provide a reference for exploring novel targeting and immunotherapy approaches for STAD.

## Figures and Tables

**Figure 1 genes-13-02214-f001:**
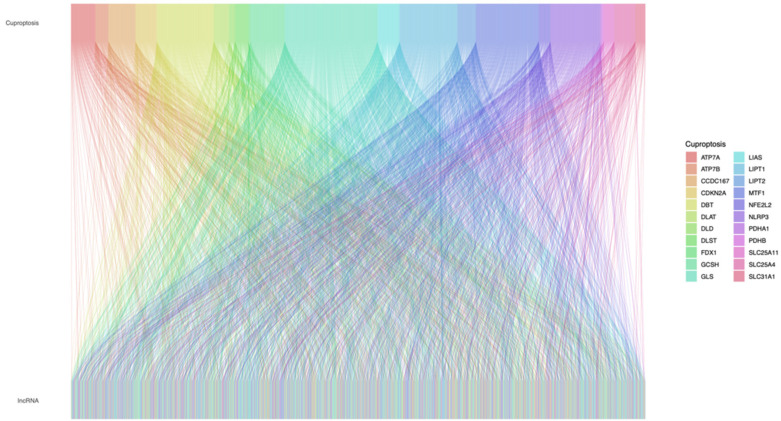
The relationships between cuproptosis-related genes and cuproptosis-related lncRNAs in the Sankey diagram. lncRNA, long noncoding RNA.

**Figure 2 genes-13-02214-f002:**
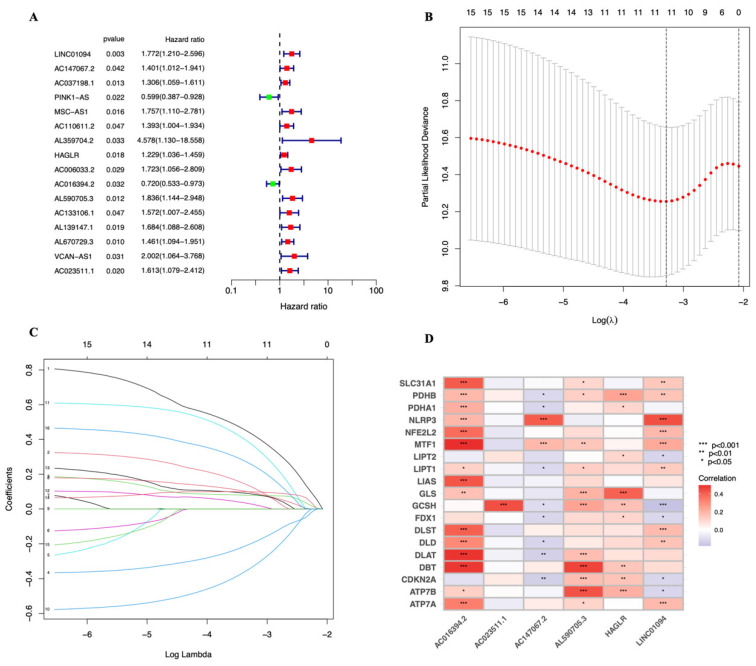
Identification of cuproptosis-associated lncRNA signature in STAD. (**A**) Univariate Cox regression analysis for identifying the prognostic cuproptosis-related lncRNAs. (**B**,**C**) Lasso–Cox regression analysis was performed to construct prognostic prediction models. Each curve in Figure 2C represents the change trajectory of each coefficient of the independent variable (representing a LncRNA in this study). The ordinate is the value of the coefficient, the lower abscissa is log (λ), and the upper abscissa represents the number of non-zero coefficients in the model at this time. (**D**) Correlation of lncRNAs with cuproptosis-related genes in risk models.

**Figure 3 genes-13-02214-f003:**
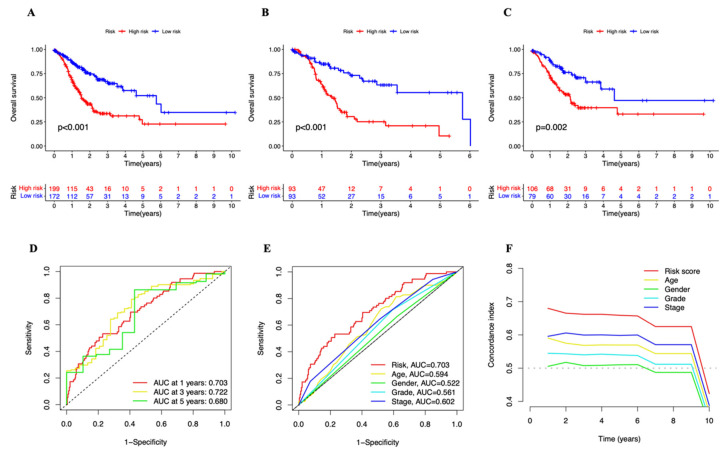
Validation of cuproptosis-associated lncRNA signature in STAD (**A**–**C**) Kaplan–Meier survival curves of overall survival of STAD patients in low- and high-risk groups in the entire, train, and test sets, respectively. (**D**) ROC curve of the cuproptosis-associated lncRNA signature in the entire set. ROC curve, receiver operating characteristics curve; AUC, area under the curve; *p* < 0.05, statistically significant. (**E**) ROC demonstrated the predictive accuracy of the risk model was superior to other clinical parameters. (**F**) The C-index curve showed the predictive accuracy of the risk model was superior to other clinical parameters. C-index curve, concordance-index curve.

**Figure 4 genes-13-02214-f004:**
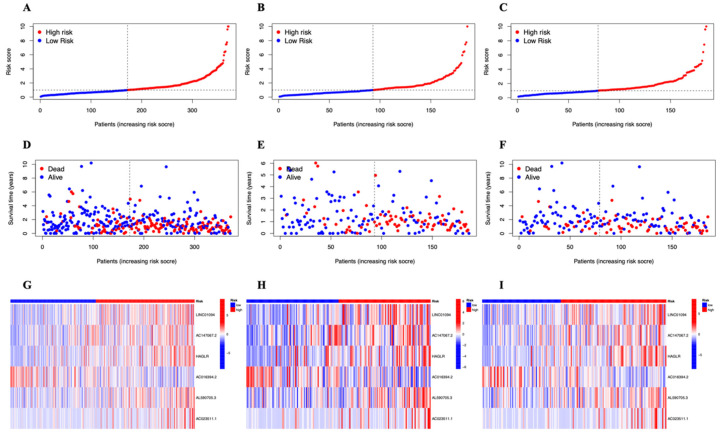
Prognosis of the risk model in different groups. (**A**–**C**) The distribution of overall survival risk scores in the entire, train, and test sets. (**D**–**F**) Survival time and survival status in the entire, train, and test sets. (**G**–**I**) Heat maps of six lncRNA expressions in the entire, train, and test sets.

**Figure 5 genes-13-02214-f005:**
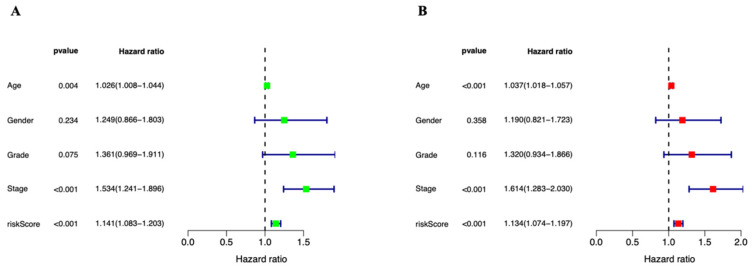
Independence of the cuproptosis-related lncRNAs prognostic marker in predicting overall survival of STAD patients. (**A**) Univariate Cox analysis. Age, stage, and risk score were statistically significant in predicting the overall survival of STAD patients. (**B**) Multivariate Cox analysis. Age, stage, and risk score were statistically significant in predicting the overall survival of STAD patients.

**Figure 6 genes-13-02214-f006:**
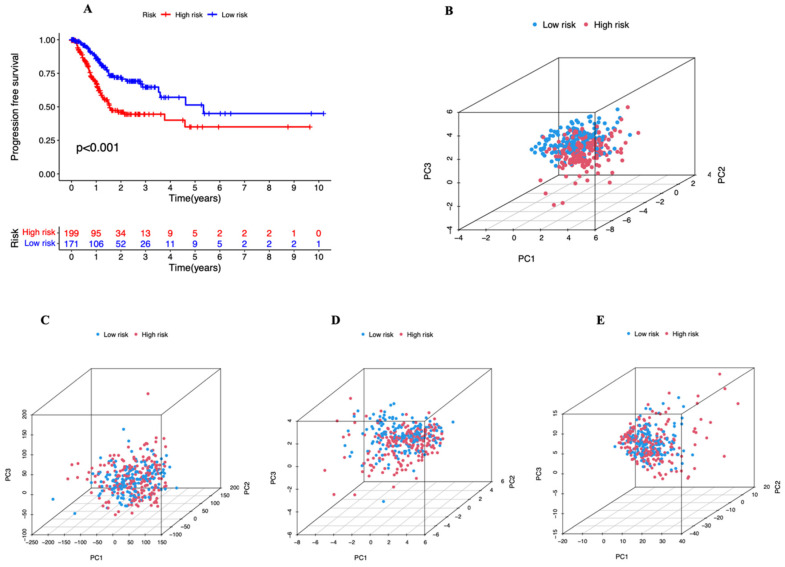
(**A**) Kaplan–Meier curves of progression-free survival (PFS) in STAD patients. (**B**–**E**) PCA in both groups of patients. (**B**) PCA of six risk lncRNAs in the prognostic marker. (**C**) PCA of all genes. (**D**) Twenty-two cuproptosis-related genes. (**E**) Sixteen cuproptosis-related lncRNAs. PCA, principal component analysis.

**Figure 7 genes-13-02214-f007:**
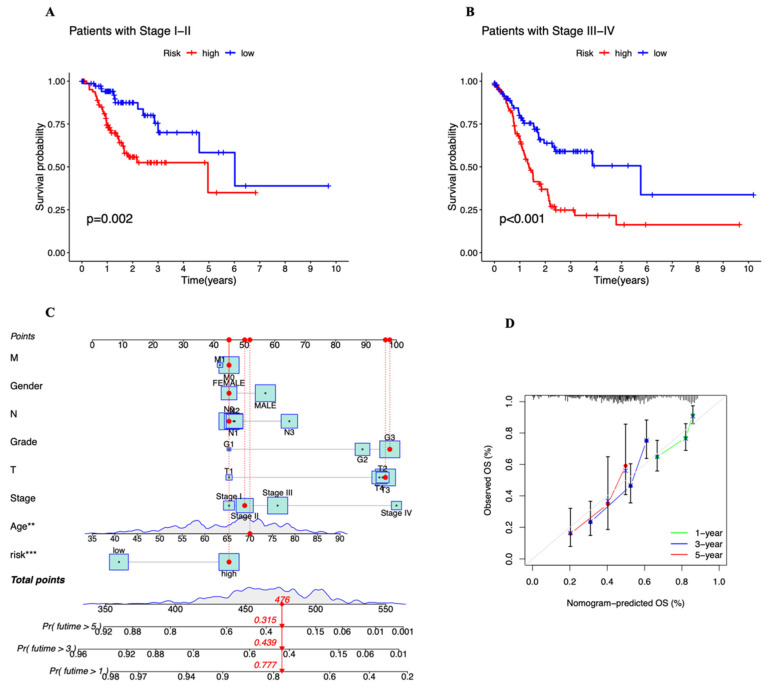
(**A**,**B**) Kaplan–Meier curves depicting subgroup survival analysis stratified by stage. *p* < 0.05, statistically significant. (**C**) The nomogram for risk score could predict the probability of survival based on the total points. (**D**) The calibration curves test the agreement between actual and predicted outcomes for STAD patients’ overall survival at 1-year (green line), 3-year (blue line), and 5-year (red line). ** *p* < 0.01, *** *p* < 0.001.

**Figure 8 genes-13-02214-f008:**
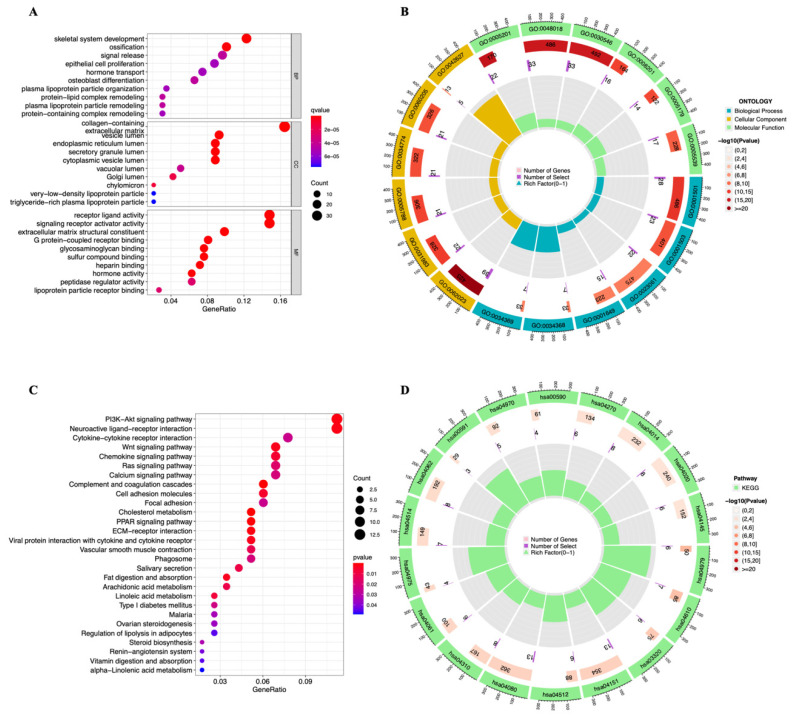
GO and KEGG analysis. (**A**,**B**) Gene ontology (GO) analysis demonstrated the richness of molecular biological processes (BP), cellular components (CC), and molecular functions (MF). (**C**,**D**) KEGG pathway analysis showed significantly enriched pathways.

**Figure 9 genes-13-02214-f009:**
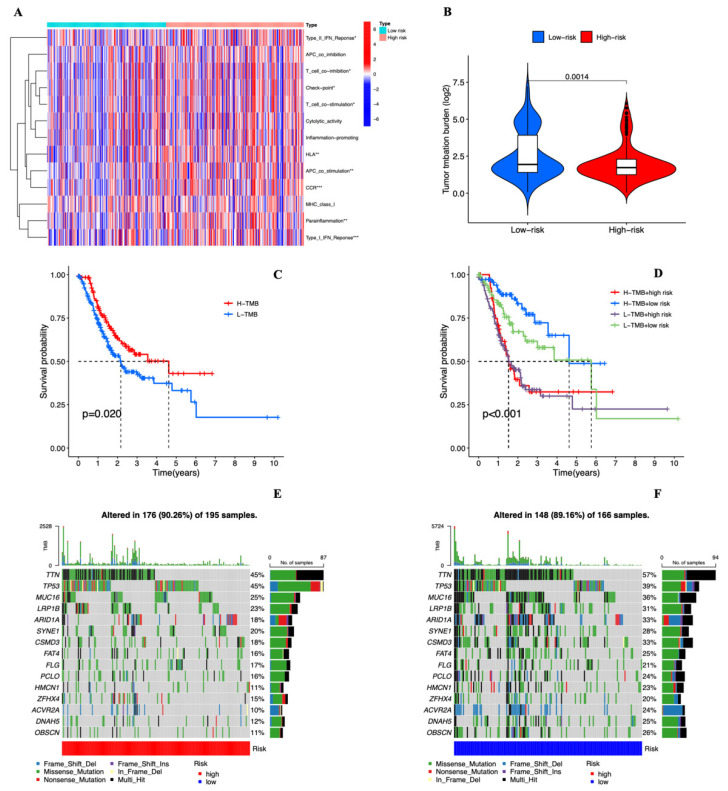
(**A**) The ssGSEA scores of immune cells and immune function in the high- and low-risk groups. “ssGSEA”, single sample gene set enrichment analysis. (**B**) The TMB in STAD patients between the high-risk and low-risk groups. TMB, tumor mutational burden; *p* < 0.05, statistically significant. (**C**) Higher TMB levels demonstrated poorer OS, *p* < 0.05, statistically significant. (**D**) Kaplan–Meier curves for patients by both risk score and TMB, *p* < 0.05, statistically significant. (**E**,**F**) Waterfall plots of somatic mutation characteristics in the high- and low-risk groups.

**Figure 10 genes-13-02214-f010:**
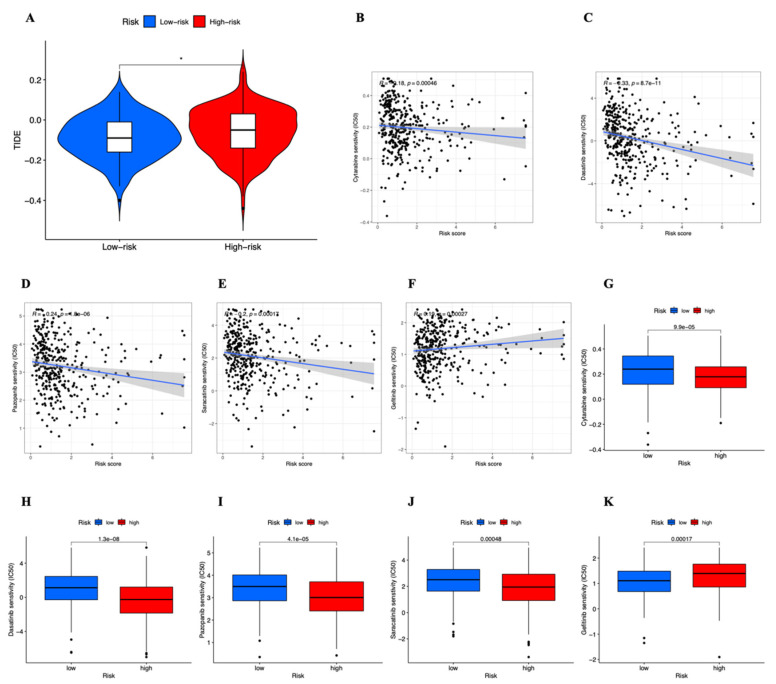
(**A**) TIDE scores between the high- and low-risk groups. (**B**–**K**) The role of risk score model in STAD treatment. The correlation between the risk score and estimated IC50 value of (**B**) Cytarabine, (**C**) Dasatinib, (**D**) Pazopanib, (**E**) Saracatinib and (**F**) Gefitinib. Comparison of estimated IC50 value of (**G**) Cytarabine, (**H**) Dasatinib, (**I**) Pazopanib, (**J**) Saracatinib and (**K**) Gefitinib between high- and low-risk groups.

**Table 1 genes-13-02214-t001:** The clinical characteristics of STAD patients in the training and test group.

Characteristics	Type	Total (%)	Test Group (%)	Train Group (%)	*p*-Value
Age	≤65	163 (43.94%)	86 (46.49%)	77 (41.4%)	0.3511
	>65	205 (55.26%)	97 (52.43%)	108 (58.06%)	
	unknow	3 (0.81%)	2 (1.08%)	1 (0.54%)	
Gender	FEMALE	133 (35.85%)	69 (37.3%)	64 (34.41%)	0.637
	MALE	238 (64.15%)	116 (62.7%)	122 (65.59%)	
Grade	G1	10 (2.7%)	3 (1.62%)	7 (3.76%)	0.2236
	G2	134 (36.12%)	62 (33.51%)	72 (38.71%)	
	G3	218 (58.76%)	115 (62.16%)	103 (55.38%)	
	unknown	9 (2.43%)	5 (2.7%)	4 (2.15%)	
Stage	Stage I	50 (13.48%)	23 (12.43%)	27 (14.52%)	0.1095
	Stage II	111 (29.92%)	50 (27.03%)	61 (32.8%)	
	Stage III	149 (40.16%)	85 (45.95%)	64 (34.41%)	
	Stage IV	38 (10.24%)	15 (8.11%)	23 (12.37%)	
	unknown	23 (6.2%)	12 (6.49%)	11 (5.91%)	
T Stage	T1	18 (4.85%)	7 (3.78%)	11 (5.91%)	0.8158
	T2	78 (21.02%)	39 (21.08%)	39 (20.97%)	
	T3	167 (45.01%)	85 (45.95%)	82 (44.09%)	
	T4	100 (26.95%)	50 (27.03%)	50 (26.88%)	
	unknown	8 (2.16%)	4 (2.16%)	4(2.15%)	
M Stage	M0	328 (88.41%)	165 (89.19%)	163(87.63%)	1
	M1	25 (6.74%)	13(7.03%)	12 (6.45%)	
	unknown	18 (4.85%)	7 (3.78%)	11 (5.91%)	
N Stage	N0	108 (29.11%)	55 (29.73%)	53 (28.49%)	0.8005
	N1	97 (26.15%)	52 (28.11%)	45 (24.19%)	
	N2	74 (19.95%)	37 (20%)	37 (19.89%)	
	N3	74 (19.95%)	34 (18.38%)	40 (21.51%)	
	unknown	18 (4.85%)	7 (3.78%)	11 (5.91%)	

## Data Availability

Corresponding authors may be contacted for article data if there is a valid reason.

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
