# Peer review of "Cuproptosis-Related lncRNA Gene Signature Establishes a Prognostic Model of Gastric Adenocarcinoma and Evaluate the Effect of Antineoplastic Drugs"

_genes, 2022, doi:10.3390/genes13122214_

Round 1

Reviewer 1 Report

I suggest the following comments which can be helpful to follow the analyses.

1- Could you explain more how PCA (principal component ananlysis) is used on page 3 please?

2- Could you explain how you define the boundary |logFC|>1 as stated on page 3 please?

3- Please give more details about TIMER,  CIBERSORT and GSEA algorithms.

4-It is not clear for me why the training and test sets are equally separated. Because typically 80% of the data is taken as training set and the remainings are taken as test sets in analyses. Alternatively 90% of the data is taken as training and 10% is used for test sets. Such an equal division of test and training set as presented in the article are not common and probably can decrease the performance in the parameter estimation stage via the training set since this calculation is done via less number of observations.

5-Could you explain how you interpret Figure 2C please?

6-Could you explain how you select the best penalty in lasso regression that is found on page 5 please?   

Author Response

Thank you very much for kindly reviewing the article! Please see the attachment for our response letter.

Here are the codes which were mentioned in the response letter:

PCA.R

library(limma) library(scatterplot3d) setwd("C:\\biowolf\\cuproptosis\\25.PCA") myPCA=function(input=null,output=null){ rt=read.table(input, header=T, sep="\t", check.names=F) rt=as.matrix(rt) rownames(rt)=rt[,1] exp=rt[,2:ncol(rt)] dimnames=list(rownames(exp),colnames(exp)) data=matrix(as.numeric(as.matrix(exp)),nrow=nrow(exp),dimnames=dimnames) data=avereps(data) data=data[rowMeans(data)>0.5,] type=sapply(strsplit(colnames(data),"\\-"),"[",4) type=sapply(strsplit(type,""),"[",1) type=gsub("2","1",type) data=t(data[,type==0]) rownames(data)=gsub("(.*?)\\-(.*?)\\-(.*?)\\-(.*?)\\-.*","\\1\\-\\2\\-\\3",rownames(data)) risk=read.table("risk.all.txt", header=T, sep="\t", row.names=1, check.names=F) sameSample=intersect(rownames(data),rownames(risk)) data=data[sameSample,] risk=risk[sameSample,] group=as.vector(risk[,"risk"]) data.class <- rownames(data) data.pca <- prcomp(data, scale. = TRUE) color=ifelse(group=="low",4,2) pcaPredict=predict(data.pca) pdf(file=output, width=7, height=7) par(oma=c(1,1,2.5,1)) s3d=scatterplot3d(pcaPredict[,1:3], pch = 16, color=color, angle=35) legend("top", legend = c("Low risk","High risk"),pch = 16, inset = -0.2, box.col="white", xpd = TRUE, horiz = TRUE,col=c(4,2)) dev.off() } myPCA(input="symbol.txt",output="PCA.allGene.pdf") myPCA(input="cuproptosisExp.txt",output="PCA.cuproptosisGene.pdf") myPCA(input="cuproptosisLncExp.txt",output="PCA.cuproptosisLncRNA.pdf") risk=read.table("risk.all.txt", header=T, sep="\t", check.names=F, row.names=1) data=risk[,3:(ncol(risk)-2)] group=as.vector(risk[,"risk"]) data.class <- rownames(data) data.pca <- prcomp(data, scale. = TRUE) color=ifelse(group=="low",4,2) pcaPredict=predict(data.pca) pdf(file="PCA.riskLnc.pdf", width=6.5, height=6) par(oma=c(1,1,2.5,1)) s3d=scatterplot3d(pcaPredict[,1:3], pch = 16, color=color, angle=35) legend("top", legend = c("Low risk","High risk"),pch = 16, inset = -0.2, box.col="white", xpd = TRUE, horiz = TRUE,col=c(4,2)) dev.off()

GO.R

library(clusterProfiler) library(org.Hs.eg.db) library(enrichplot) library(ggplot2) library(circlize) library(RColorBrewer) library(dplyr) library("ggpubr") library(ComplexHeatmap) pvalueFilter=0.05 qvalueFilter=0.05 colorSel="qvalue" if(qvalueFilter>0.05){ colorSel="pvalue" } setwd("C:\\biowolf\\cuproptosis\\27.GO") rt=read.table("riskDiff.txt", header=T, sep="\t", check.names=F) genes=unique(as.vector(rt[,1])) entrezIDs=mget(genes, org.Hs.egSYMBOL2EG, ifnotfound=NA) entrezIDs=as.character(entrezIDs) gene=entrezIDs[entrezIDs!="NA"] kk=enrichGO(gene=gene, OrgDb=org.Hs.eg.db, pvalueCutoff=1, qvalueCutoff=1, ont="all", readable=T) GO=as.data.frame(kk) GO=GO[(GO$pvalue<pvalueFilter & GO$qvalue<qvalueFilter),] write.table(GO, file="GO.txt", sep="\t", quote=F, row.names = F) showNum=10 if(nrow(GO)<30){ showNum=nrow(GO) } pdf(file="barplot.pdf", width=8, height=7) bar=barplot(kk, drop=TRUE, showCategory=showNum, label_format=30, split="ONTOLOGY", color=colorSel) + facet_grid(ONTOLOGY~., scale='free') print(bar) dev.off() pdf(file="bubble.pdf", width=8, height=7) bub=dotplot(kk, showCategory=showNum, orderBy="GeneRatio", label_format=30, split="ONTOLOGY", color=colorSel) + facet_grid(ONTOLOGY~., scale='free') print(bub) dev.off() data=GO %>% group_by(ONTOLOGY) %>% slice_head(n=10) pdf(file="barplot.color.pdf", width=8, height=6.5) ggbarplot(data, x="Description", y="Count", fill = "ONTOLOGY", color = "white", xlab="Term", orientation = "horiz", palette = "aaas", legend = "right", sort.val = "asc", sort.by.groups=TRUE)+ scale_y_continuous(expand=c(0, 0)) + scale_x_discrete(expand=c(0,0)) dev.off() ontology.col=c("#00AFBB", "#E7B800", "#90EE90") data=GO[order(GO$p.adjust),] datasig=data[data$p.adjust<0.05,,drop=F] BP = datasig[datasig$ONTOLOGY=="BP",,drop=F] CC = datasig[datasig$ONTOLOGY=="CC",,drop=F] MF = datasig[datasig$ONTOLOGY=="MF",,drop=F] BP = head(BP,6) CC = head(CC,6) MF = head(MF,6) data = rbind(BP,CC,MF) main.col = ontology.col[as.numeric(as.factor(data$ONTOLOGY))] BgGene = as.numeric(sapply(strsplit(data$BgRatio,"/"),'[',1)) Gene = as.numeric(sapply(strsplit(data$GeneRatio,'/'),'[',1)) ratio = Gene/BgGene logpvalue = -log(data$pvalue,10) logpvalue.col = brewer.pal(n = 8, name = "Reds") f = colorRamp2(breaks = c(0,2,4,6,8,10,15,20), colors = logpvalue.col) BgGene.col = f(logpvalue) df = data.frame(GO=data$ID,start=1,end=max(BgGene)) rownames(df) = df$GO bed2 = data.frame(GO=data$ID,start=1,end=BgGene,BgGene=BgGene,BgGene.col=BgGene.col) bed3 = data.frame(GO=data$ID,start=1,end=Gene,BgGene=Gene) bed4 = data.frame(GO=data$ID,start=1,end=max(BgGene),ratio=ratio,col=main.col) bed4$ratio = bed4$ratio/max(bed4$ratio)*9.5 pdf("GO.circlize.pdf",width=10,height=10) par(omi=c(0.1,0.1,0.1,1.5)) circos.par(track.margin=c(0.01,0.01)) circos.genomicInitialize(df,plotType="none") circos.trackPlotRegion(ylim = c(0, 1), panel.fun = function(x, y) { sector.index = get.cell.meta.data("sector.index") xlim = get.cell.meta.data("xlim") ylim = get.cell.meta.data("ylim") circos.text(mean(xlim), mean(ylim), sector.index, cex = 0.8, facing = "bending.inside", niceFacing = TRUE) }, track.height = 0.08, bg.border = NA,bg.col = main.col) for(si in get.all.sector.index()) { circos.axis(h = "top", labels.cex = 0.6, sector.index = si,track.index = 1, major.at=seq(0,max(BgGene),by=100),labels.facing = "clockwise") } f = colorRamp2(breaks = c(-1, 0, 1), colors = c("green", "black", "red")) circos.genomicTrack(bed2, ylim = c(0, 1),track.height = 0.1,bg.border="white", panel.fun = function(region, value, ...) { i = getI(...) circos.genomicRect(region, value, ytop = 0, ybottom = 1, col = value[,2], border = NA, ...) circos.genomicText(region, value, y = 0.4, labels = value[,1], adj=0,cex=0.8,...) }) circos.genomicTrack(bed3, ylim = c(0, 1),track.height = 0.1,bg.border="white", panel.fun = function(region, value, ...) { i = getI(...) circos.genomicRect(region, value, ytop = 0, ybottom = 1, col = '#BA55D3', border = NA, ...) circos.genomicText(region, value, y = 0.4, labels = value[,1], cex=0.9,adj=0,...) }) circos.genomicTrack(bed4, ylim = c(0, 10),track.height = 0.35,bg.border="white",bg.col="grey90", panel.fun = function(region, value, ...) { cell.xlim = get.cell.meta.data("cell.xlim") cell.ylim = get.cell.meta.data("cell.ylim") for(j in 1:9) { y = cell.ylim[1] + (cell.ylim[2]-cell.ylim[1])/10*j circos.lines(cell.xlim, c(y, y), col = "#FFFFFF", lwd = 0.3) } circos.genomicRect(region, value, ytop = 0, ybottom = value[,1], col = value[,2], border = NA, ...) #circos.genomicText(region, value, y = 0.3, labels = value[,1], ...) }) circos.clear() middle.legend = Legend( labels = c('Number of Genes','Number of Select','Rich Factor(0-1)'), type="points",pch=c(15,15,17),legend_gp = gpar(col=c('pink','#BA55D3',ontology.col[1])), title="",nrow=3,size= unit(3, "mm") ) circle_size = unit(1, "snpc") draw(middle.legend,x=circle_size*0.42) main.legend = Legend( labels = c("Biological Process","Cellular Component", "Molecular Function"), type="points",pch=15, legend_gp = gpar(col=ontology.col), title_position = "topcenter", title = "ONTOLOGY", nrow = 3,size = unit(3, "mm"),grid_height = unit(5, "mm"), grid_width = unit(5, "mm") ) logp.legend = Legend( labels=c('(0,2]','(2,4]','(4,6]','(6,8]','(8,10]','(10,15]','(15,20]','>=20'), type="points",pch=16,legend_gp=gpar(col=logpvalue.col),title="-log10(Pvalue)", title_position = "topcenter",grid_height = unit(5, "mm"),grid_width = unit(5, "mm"), size = unit(3, "mm") ) lgd = packLegend(main.legend,logp.legend) circle_size = unit(1, "snpc") print(circle_size) draw(lgd, x = circle_size*0.85, y=circle_size*0.55,just = "left") dev.off()

KEGG.R

library(clusterProfiler) library(org.Hs.eg.db) library(enrichplot) library(ggplot2) library(circlize) library(RColorBrewer) library(dplyr) library(ComplexHeatmap) pvalueFilter=0.05 qvalueFilter=0.05 colorSel="qvalue" if(qvalueFilter>0.05){ colorSel="pvalue" } setwd("C:\\biowolf\\cuproptosis\\28.KEGG") rt=read.table("riskDiff.txt", header=T, sep="\t", check.names=F) genes=unique(as.vector(rt[,1])) entrezIDs=mget(genes, org.Hs.egSYMBOL2EG, ifnotfound=NA) entrezIDs=as.character(entrezIDs) rt=data.frame(genes, entrezID=entrezIDs) gene=entrezIDs[entrezIDs!="NA"] kk <- enrichKEGG(gene=gene, organism="hsa", pvalueCutoff=1, qvalueCutoff=1) KEGG=as.data.frame(kk) KEGG$geneID=as.character(sapply(KEGG$geneID,function(x)paste(rt$genes[match(strsplit(x,"/")[[1]],as.character(rt$entrezID))],collapse="/"))) KEGG=KEGG[(KEGG$pvalue<pvalueFilter & KEGG$qvalue<qvalueFilter),] write.table(KEGG, file="KEGG.txt", sep="\t", quote=F, row.names = F) showNum=30 if(nrow(KEGG)<showNum){ showNum=nrow(KEGG) } pdf(file="barplot.pdf", width=9, height=7) barplot(kk, drop=TRUE, showCategory=showNum, label_format=130, color=colorSel) dev.off() pdf(file="bubble.pdf", width = 9, height = 7) dotplot(kk, showCategory=showNum, orderBy="GeneRatio", label_format=130, color=colorSel) dev.off() Pathway.col=c("#90EE90", "#E7B800", "#00AFBB") showNum=18 data=KEGG[order(KEGG$p.adjust),] if(nrow(KEGG)>showNum){ data=data[1:showNum,] } data$Pathway="KEGG" main.col = Pathway.col[as.numeric(as.factor(data$Pathway))] BgGene = as.numeric(sapply(strsplit(data$BgRatio,"/"),'[',1)) Gene = as.numeric(sapply(strsplit(data$GeneRatio,'/'),'[',1)) ratio = Gene/BgGene logpvalue = -log(data$pvalue,10) logpvalue.col = brewer.pal(n = 8, name = "Reds") f = colorRamp2(breaks = c(0,2,4,6,8,10,15,20), colors = logpvalue.col) BgGene.col = f(logpvalue) df = data.frame(KEGG=data$ID,start=1,end=max(BgGene)) rownames(df) = df$KEGG bed2 = data.frame(KEGG=data$ID,start=1,end=BgGene,BgGene=BgGene,BgGene.col=BgGene.col) bed3 = data.frame(KEGG=data$ID,start=1,end=Gene,BgGene=Gene) bed4 = data.frame(KEGG=data$ID,start=1,end=max(BgGene),ratio=ratio,col=main.col) bed4$ratio = bed4$ratio/max(bed4$ratio)*9.5 pdf(file="KEGG.circlize.pdf",width=10,height=10) par(omi=c(0.1,0.1,0.1,1.5)) circos.par(track.margin=c(0.01,0.01)) circos.genomicInitialize(df,plotType="none") circos.trackPlotRegion(ylim = c(0, 1), panel.fun = function(x, y) { sector.index = get.cell.meta.data("sector.index") xlim = get.cell.meta.data("xlim") ylim = get.cell.meta.data("ylim") circos.text(mean(xlim), mean(ylim), sector.index, cex = 0.8, facing = "bending.inside", niceFacing = TRUE) }, track.height = 0.08, bg.border = NA,bg.col = main.col) for(si in get.all.sector.index()) { circos.axis(h = "top", labels.cex = 0.6, sector.index = si,track.index = 1, major.at=seq(0,max(BgGene),by=100),labels.facing = "clockwise") } f = colorRamp2(breaks = c(-1, 0, 1), colors = c("green", "black", "red")) circos.genomicTrack(bed2, ylim = c(0, 1),track.height = 0.1,bg.border="white", panel.fun = function(region, value, ...) { i = getI(...) circos.genomicRect(region, value, ytop = 0, ybottom = 1, col = value[,2], border = NA, ...) circos.genomicText(region, value, y = 0.4, labels = value[,1], adj=0,cex=0.8,...) }) circos.genomicTrack(bed3, ylim = c(0, 1),track.height = 0.1,bg.border="white", panel.fun = function(region, value, ...) { i = getI(...) circos.genomicRect(region, value, ytop = 0, ybottom = 1, col = '#BA55D3', border = NA, ...) circos.genomicText(region, value, y = 0.4, labels = value[,1], cex=0.9,adj=0,...) }) circos.genomicTrack(bed4, ylim = c(0, 10),track.height = 0.35,bg.border="white",bg.col="grey90", panel.fun = function(region, value, ...) { cell.xlim = get.cell.meta.data("cell.xlim") cell.ylim = get.cell.meta.data("cell.ylim") for(j in 1:9) { y = cell.ylim[1] + (cell.ylim[2]-cell.ylim[1])/10*j circos.lines(cell.xlim, c(y, y), col = "#FFFFFF", lwd = 0.3) } circos.genomicRect(region, value, ytop = 0, ybottom = value[,1], col = value[,2], border = NA, ...) #circos.genomicText(region, value, y = 0.3, labels = value[,1], ...) }) circos.clear() #????Ȧͼ?м???ͼ?? middle.legend = Legend( labels = c('Number of Genes','Number of Select','Rich Factor(0-1)'), type="points",pch=c(15,15,17),legend_gp = gpar(col=c('pink','#BA55D3',Pathway.col[1])), title="",nrow=3,size= unit(3, "mm") ) circle_size = unit(1, "snpc") draw(middle.legend,x=circle_size*0.42) #????KEGG??????ͼ?? main.legend = Legend( labels = c("KEGG"), type="points",pch=15, legend_gp = gpar(col=Pathway.col), title_position = "topcenter", title = "Pathway", nrow = 3,size = unit(3, "mm"),grid_height = unit(5, "mm"), grid_width = unit(5, "mm") ) #????pvalue??ͼ?? logp.legend = Legend( labels=c('(0,2]','(2,4]','(4,6]','(6,8]','(8,10]','(10,15]','(15,20]','>=20'), type="points",pch=16,legend_gp=gpar(col=logpvalue.col),title="-log10(Pvalue)", title_position = "topcenter",grid_height = unit(5, "mm"),grid_width = unit(5, "mm"), size = unit(3, "mm") ) lgd = packLegend(main.legend,logp.legend) circle_size = unit(1, "snpc") print(circle_size) draw(lgd, x = circle_size*0.85, y=circle_size*0.55,just = "left") dev.off()

immuneFunction.R

library(limma) library(GSVA) library(GSEABase) library(pheatmap) library(reshape2) expFile="symbol.txt" gmtFile="immune.gmt" riskFile="risk.all.txt" setwd("C:\\biowolf\\cuproptosis\\29.immFunction") rt=read.table(expFile, header=T, sep="\t", check.names=F) rt=as.matrix(rt) rownames(rt)=rt[,1] exp=rt[,2:ncol(rt)] dimnames=list(rownames(exp),colnames(exp)) mat=matrix(as.numeric(as.matrix(exp)),nrow=nrow(exp),dimnames=dimnames) mat=avereps(mat) mat=mat[rowMeans(mat)>0,] geneSet=getGmt(gmtFile, geneIdType=SymbolIdentifier()) ssgseaScore=gsva(mat, geneSet, method='ssgsea', kcdf='Gaussian', abs.ranking=TRUE) normalize=function(x){ return((x-min(x))/(max(x)-min(x)))} data=normalize(ssgseaScore) ssgseaOut=rbind(id=colnames(data), data) write.table(ssgseaOut, file="immFunScore.txt", sep="\t", quote=F, col.names=F) group=sapply(strsplit(colnames(data),"\\-"), "[", 4) group=sapply(strsplit(group,""), "[", 1) group=gsub("2", "1", group) data=t(data[,group==0]) rownames(data)=gsub("(.*?)\\-(.*?)\\-(.*?)\\-(.*?)\\-.*", "\\1\\-\\2\\-\\3", rownames(data)) data=t(avereps(data)) risk=read.table(riskFile, header=T, sep="\t", check.names=F, row.names=1) lowSample=row.names(risk[risk$risk=="low",]) highSample=row.names(risk[risk$risk=="high",]) lowData=data[,lowSample] highData=data[,highSample] data=cbind(lowData, highData) conNum=ncol(lowData) treatNum=ncol(highData) sampleType=c(rep(1,conNum), rep(2,treatNum)) sigVec=c() for(i in row.names(data)){ test=wilcox.test(data[i,] ~ sampleType) pvalue=test$p.value Sig=ifelse(pvalue<0.001,"***",ifelse(pvalue<0.01,"**",ifelse(pvalue<0.05,"*",""))) sigVec=c(sigVec, paste0(i, Sig)) } row.names(data)=sigVec Type=c(rep("Low risk",conNum), rep("High risk",treatNum)) Type=factor(Type, levels=c("Low risk", "High risk")) names(Type)=colnames(data) Type=as.data.frame(Type) pdf("heatmap.pdf", width=8, height=4.6) pheatmap(data, annotation=Type, color = colorRampPalette(c(rep("blue",5), "white", rep("red",5)))(100), cluster_cols =F, cluster_rows =T, scale="row", show_colnames=F, show_rownames=T, fontsize=7, fontsize_row=7, fontsize_col=7) dev.off()

Lasso Regression and Model Building .R

library(survival) library(caret) library(glmnet) library(survminer) library(timeROC) coxPfilter=0.05 setwd("C:\\biowolf\\cuproptosis\\14.model") rt=read.table("expTime.txt", header=T, sep="\t", check.names=F, row.names=1) rt$futime[rt$futime<=0]=1 rt$futime=rt$futime/365 rt[,3:ncol(rt)]=log2(rt[,3:ncol(rt)]+1) bioForest=function(coxFile=null,forestFile=null,forestCol=null){ rt <- read.table(coxFile,header=T,sep="\t",row.names=1,check.names=F) gene <- rownames(rt) hr <- sprintf("%.3f",rt$"HR") hrLow <- sprintf("%.3f",rt$"HR.95L") hrHigh <- sprintf("%.3f",rt$"HR.95H") Hazard.ratio <- paste0(hr,"(",hrLow,"-",hrHigh,")") pVal <- ifelse(rt$pvalue<0.001, "<0.001", sprintf("%.3f", rt$pvalue)) pdf(file=forestFile, width=7, height=6) n <- nrow(rt) nRow <- n+1 ylim <- c(1,nRow) layout(matrix(c(1,2),nc=2),width=c(3,2.5)) xlim = c(0,3) par(mar=c(4,2.5,2,1)) plot(1,xlim=xlim,ylim=ylim,type="n",axes=F,xlab="",ylab="") text.cex=0.8 text(0,n:1,gene,adj=0,cex=text.cex) text(1.5-0.5*0.2,n:1,pVal,adj=1,cex=text.cex);text(1.5-0.5*0.2,n+1,'pvalue',cex=text.cex,adj=1) text(3,n:1,Hazard.ratio,adj=1,cex=text.cex);text(3,n+1,'Hazard ratio',cex=text.cex,adj=1,) par(mar=c(4,1,2,1),mgp=c(2,0.5,0)) LOGindex = 10 hrLow = log(as.numeric(hrLow),LOGindex) hrHigh = log(as.numeric(hrHigh),LOGindex) hr = log(as.numeric(hr),LOGindex) xlim = c(floor(min(hrLow,hrHigh)),ceiling(max(hrLow,hrHigh))) plot(1,xlim=xlim,ylim=ylim,type="n",axes=F,ylab="",xaxs="i",xlab="Hazard ratio") arrows(as.numeric(hrLow),n:1,as.numeric(hrHigh),n:1,angle=90,code=3,length=0.05,col="darkblue",lwd=2.5) abline(v=log(1,LOGindex),col="black",lty=2,lwd=2) boxcolor = ifelse(as.numeric(hr) > log(1,LOGindex), forestCol[1],forestCol[2]) points(as.numeric(hr), n:1, pch = 15, col = boxcolor, cex=1.3) a1 = axis(1,labels=F,tick=F) axis(1,a1,10^a1) dev.off() } n=1 for(i in 1:n){ inTrain<-createDataPartition(y=rt[,2], p=0.5, list=F) train<-rt[inTrain,] test<-rt[-inTrain,] trainOut=cbind(id=row.names(train),train) testOut=cbind(id=row.names(test),test) outUniTab=data.frame() sigGenes=c("futime","fustat") for(i in colnames(train[,3:ncol(train)])){ cox <- coxph(Surv(futime, fustat) ~ train[,i], data = train) coxSummary = summary(cox) coxP=coxSummary$coefficients[,"Pr(>|z|)"] if(coxP<coxPfilter){ sigGenes=c(sigGenes,i) outUniTab=rbind(outUniTab, cbind(id=i, HR=coxSummary$conf.int[,"exp(coef)"], HR.95L=coxSummary$conf.int[,"lower .95"], HR.95H=coxSummary$conf.int[,"upper .95"], pvalue=coxSummary$coefficients[,"Pr(>|z|)"]) ) } } uniSigExp=train[,sigGenes] uniSigExpOut=cbind(id=row.names(uniSigExp),uniSigExp) if(ncol(uniSigExp)<6){next} #lasso Regression x=as.matrix(uniSigExp[,c(3:ncol(uniSigExp))]) y=data.matrix(Surv(uniSigExp$futime,uniSigExp$fustat)) fit <- glmnet(x, y, family = "cox", maxit = 1000) cvfit <- cv.glmnet(x, y, family="cox", maxit = 1000) coef <- coef(fit, s = cvfit$lambda.min) index <- which(coef != 0) actCoef <- coef[index] lassoGene=row.names(coef)[index] lassoSigExp=uniSigExp[,c("futime", "fustat", lassoGene)] lassoSigExpOut=cbind(id=row.names(lassoSigExp), lassoSigExp) geneCoef=cbind(Gene=lassoGene, Coef=actCoef) if(nrow(geneCoef)<2){next} multiCox <- coxph(Surv(futime, fustat) ~ ., data = lassoSigExp) multiCox=step(multiCox, direction = "both") multiCoxSum=summary(multiCox) outMultiTab=data.frame() outMultiTab=cbind( coef=multiCoxSum$coefficients[,"coef"], HR=multiCoxSum$conf.int[,"exp(coef)"], HR.95L=multiCoxSum$conf.int[,"lower .95"], HR.95H=multiCoxSum$conf.int[,"upper .95"], pvalue=multiCoxSum$coefficients[,"Pr(>|z|)"]) outMultiTab=cbind(id=row.names(outMultiTab),outMultiTab) outMultiTab=outMultiTab[,1:2] riskScore=predict(multiCox,type="risk",newdata=train) coxGene=rownames(multiCoxSum$coefficients) coxGene=gsub("`","",coxGene) outCol=c("futime","fustat",coxGene) medianTrainRisk=median(riskScore) risk=as.vector(ifelse(riskScore>medianTrainRisk,"high","low")) trainRiskOut=cbind(id=rownames(cbind(train[,outCol],riskScore,risk)),cbind(train[,outCol],riskScore,risk)) riskScoreTest=predict(multiCox,type="risk",newdata=test) riskTest=as.vector(ifelse(riskScoreTest>medianTrainRisk,"high","low")) testRiskOut=cbind(id=rownames(cbind(test[,outCol],riskScoreTest,riskTest)),cbind(test[,outCol],riskScore=riskScoreTest,risk=riskTest)) diff=survdiff(Surv(futime, fustat) ~risk,data = train) pValue=1-pchisq(diff$chisq, df=1) diffTest=survdiff(Surv(futime, fustat) ~riskTest,data = test) pValueTest=1-pchisq(diffTest$chisq, df=1) predictTime=1 roc=timeROC(T=train$futime, delta=train$fustat, marker=riskScore, cause=1, times=c(predictTime), ROC=TRUE) rocTest=timeROC(T=test$futime, delta=test$fustat, marker=riskScoreTest, cause=1, times=c(predictTime), ROC=TRUE) if((pValue<0.01) & (roc$AUC[2]>0.68) & (pValueTest<0.02) & (rocTest$AUC[2]>0.65)){ write.table(trainOut,file="data.train.txt",sep="\t",quote=F,row.names=F) write.table(testOut,file="data.test.txt",sep="\t",quote=F,row.names=F) write.table(outUniTab,file="uni.trainCox.txt",sep="\t",row.names=F,quote=F) write.table(uniSigExpOut,file="uni.SigExp.txt",sep="\t",row.names=F,quote=F) bioForest(coxFile="uni.trainCox.txt",forestFile="uni.foreast.pdf",forestCol=c("red","green")) #lasso write.table(lassoSigExpOut,file="lasso.SigExp.txt",sep="\t",row.names=F,quote=F) pdf("lasso.lambda.pdf") plot(fit, xvar = "lambda", label = TRUE) dev.off() pdf("lasso.cvfit.pdf") plot(cvfit) abline(v=log(c(cvfit$lambda.min,cvfit$lambda.1se)), lty="dashed") dev.off() write.table(outMultiTab,file="multiCox.txt",sep="\t",row.names=F,quote=F) write.table(trainRiskOut,file="risk.train.txt",sep="\t",quote=F,row.names=F) write.table(testRiskOut,file="risk.test.txt",sep="\t",quote=F,row.names=F) allRiskOut=rbind(trainRiskOut, testRiskOut) write.table(allRiskOut,file="risk.all.txt",sep="\t",quote=F,row.names=F) break } }

Reviewer 2 Report

 figures,tables and images were well designed.

Author Response

Thank you very much for reviewing the article! Please see the attachment for response letter.
